# Assessing port service quality: An application of the extension fuzzy AHP and importance-performance analysis

**Thang Quyet Nguyen** [1], **Lan Thi Tuyet Ngo**[2], **Nguyen Tan Huynh** [3,4] *, **Thanh Le Quoc**[5,6], **Long Van Hoang**[7]

**1** Faculty of Tourism & Hospitality Management, HUTECH University, Ho Chi Minh city, Vietnam, **2** Post Graduate Department, Dong Nai Technology University, Dong Nai, Vietnam, **3** Faculty of Economics and Management, Dong Nai Technology University, Bien Hoa, Dong Nai, Vietnam, **4** Department of Shipping & Transportation Management, National Kaohsiung University of Science and Technology, Taiwan, R.O.C, **5** Graduate School, University of Finance-Marketing, Ho Chi Minh city, Vietnam, **6** PhD Candidate, Universite ParisSaclay, Univ Evry, IMT-BS, LITEM, 91025, Paris, Evry-Courouronnes, Frances., FR., **7** Faculty of Management, Ho Chi Minh City University of Law, Ho Chi Minh city 700000, Vietnam

☙ These authors contributed equally to this work.
* i108189105@nkust.edu.tw

## Abstract

It is argued that ports are playing a crucial role in developing nations' economy. Still, solutions to improving port service quality (PSQ) to boost ports' competitive capacity is questionable. Hence, this study aims to investigate port service quality (PSQ) by using integration of the extension Fuzzy Analytic Hierarchy Process and Importance-Performance Analysis (IPA) from port users' perspectives. From the relevant literature and expert interview, the hierarchical structure of PSQ embracing six dimensions with 29 criteria was first established. To test the research model, the Dong Nai port joint stock company (DNPC) and their port-service users were empirically investigated. It is found that: (1) the importance degree of dimensions is ranked as follow: empathy (21.07%), tangibles (20.15%), assurance (15.97%), reliability (15.54%), responsiveness (12.53%), diversity (14.74%); (2) for criteria of PSQ, top five criteria concerned by shipping companies and ocean freight forwarders comprise: "proactive provision of vessel schedules", "cargo handling facilities and equipment", "detailed schedule", "accuracy and consistency of schedules", and "geographical location"; (3) there are four service attributes (SAs) needing to prioritize for improvement, including "perfect transportation of cargos", "ability in dealing with cargo damage", "willingness in helping customers", "provision of special cargo-related services". The practical policy is that port authorities should transfer the limited resources from SAs in Quadrant IV to Quadrant II to enhance the PSQ.

## 1. Introduction

With globalization and the progressive development of logistics and supply chains, container ports are playing a very important role in the economic growth of nations, especially with long coast line. They are also the main gateway for foreign markets [1]; thereby, the enhancement of port service quality (PSQ) is considered the main benchmark of a nation's competitive

**Funding:** The author(s) received no specific funding for this work.

**Competing interests:** The authors have declared that no competing interests exist.

capacity [2]. Moreover, the progressive development of container throughput, as well as the increasingly trend of globalization in the port industry has been leading to fierce competition among port companies (PCs) in attracting port-service users, including shipping liners, shippers, and maritime freight forwarders [3–5]. Thus, port service quality (PSQ) is a key issue concerning PCs because it affects port-users' choice for container ports and terminals, thereby influencing the business efficiency of PCs. However, most present research focused on determinants of port choice factors [6, 7], and the assessment of the investment environment of seaports [8–10]. By contrast, PSQ-related studies and how to improve PSQ are lacking.

Also, port privatization is considered as one of the main factors leading to the port competition nowadays [11]. It is explained that the birth of private ports has resulted in the competitive pressure for state-owned ports ever than before [12]. Furthermore, the function of ports is being expanded, from a part of maritime transportation [9] to the integration in the global traffic and the logistic system [13]. This encourages port authorities to seek effective solutions to boost the competitive advantages and maintain the market share. Hence, the most important thing is that PCs must differentiate themselves by using long-term strategies to get ahead of their competitors in business operations. That is why identification of determinants of PSQ enable PCs to advance competitiveness and gain profits sustainably in the dynamic business environment [5, 14, 15].

Although the prior research relatively succeed in developing the measurement scale of PSQ [16], assessing the performance efficiency of ports [15, 17], ranking the weights of PSQ dimensions [5, 14], their main limitation is that none of them yield insights to explore which service attributes (SAs) that PCs should improve to meet customer satisfaction. Further, the assessment of PSQ can be considered as a problem of multiple-criteria decision analysis (MCDA). It is posited that there are many various algorithms regarding the MCDA approach, such as Technique for Order Preference by Similarity to Ideal Solution (TOPSIS) [18], the weighted aggregated sum product assessment (WASPAS) [19], the cross-impact matrix multiplication applied to classification (MICMAC) [20], the Vlsekriterijumska Optimizacija I Lompromisno Resenje (VIKOR) [21], especially the Analytic Hierarchy Process (AHP), which can be seen as the most well-known in the relevant literature [18, 22]. On top of that, Wang, Dang [19] emphasized that exact numbers (or crisp numbers) cannot capture uncertain and imperfect human ratings in many real-world situations. Thus, triangular fuzzy numbers (TFNs) are argued to be an attractive alternative for assessing qualitative factors.

To address the literature gap, the main purpose of this study is to assess PSQ by using the extension Fuzzy Analytic Hierarchy Process (F-AHP) and Importance-Performance Analysis (IPA) from port users' perspectives. In this study, dimensions of PSQ initially identified based on relevant literature and qualitative approach. Next, their degree of importance and satisfaction were weighed by F-AHP with Fuzzy Triangular Numbers (FTNs). Then, the IPA model was finally employed to confirm which SAs need to provide priorities for allocating limited resources. For an empirical study, companies using port service provided by the Dong Nai port joint stock company (DNPC) were investigated to validate the research model.

This research proceeds as follows: Section 2 briefly introduces literature reviews; Section 3 describes the research method. Section 4 represents results, discussions, and managerial implications of the empirical study. Finally, the main conclusions and several limitations are summarized in the last section of this article.

## 2. Literature review

### 2.1. Theory of the fuzzy set

The fuzzy set (FS) originally defined by Zadeh [23], is a collection of real numbers having partial membership in the set [24]. Unlike numbers in the crisp set which can be true or false,

nothing in between, an element in FS may belong to a set or not. Thus, not only is the FS used to cope with equivocation [25], imprecision [26], uncertainties [27], and ambiguity [28] in decision-making, but also provides flexible and emotional solutions to establish potential interference networks in solving complex control and classification problems [29]. Let $T$ be the domain of discourse and t be its elements. According to the theory of the FS, FS B of the domain T is defined by the function, $\mu_B(t)$, as follow $\mu_B(t):T->[0,1]$,

where:

$$\mu_B(t) = \begin{cases} 1 & \text{if } t \in B \\ 0 & \text{if } t \notin B \\ u & \text{if } t \text{ is partially in } B, \ (0 < u < 1) \end{cases}$$

Thus, $\mu_B(t)$ is called as the membership function (MF) of FS B while the value of $\mu_B(t)$ represents the degree of membership, which is also the member value of the element t in set B.

In present, FS is applied to many different sectors and also viewed to be more effective in assessing human's subjective judgments than the Likert scale [30–35]. García-Dastugue and Eroglu [36] explored that service quality of hospitals of Italia included four main constructs (healthcare staff and doctors, responsiveness, relationships between patients and doctors, and additional services) and 15 pertinent criteria. Sirisawat and Kiatcharoenpol [8] and Prakash and Barua [37] utilized the hybrid methods of fuzzy AHP and fuzzy TOPSIS to rank solutions to solve barriers for reverse logistics practices in manufacturing industries of Thailand and India. Furthermore, the theory of the FS is used for analyzing the internal environment of tourism and hospitality, for selecting the service providers in transportation, for determining the efficiency of educational units, and for investigating service quality in other service sectors (Table 1).

## 2.2. The determinants of PSQ

PSQ is viewed as a scale of how well the port service provided satisfies users' expectations, regardless of whether the specification of the latter is beforehand or not [15, 17, 50]. Many recent studies have applied the SERQUAL scale to explore the service quality, customer satisfaction and customer retention in different sectors, namely tourism [51], logistics [36], healthcare [38], marketing [52], e-commerce [53], e-retailing [54], banking service [55]. Only a few port service-related studies have been carried out so far.

**Table 1. The prior studies use the theory of the FS.**

| Authors | Research areas |
|---|---|
| García-Dastugue and Eroglu [36], Singh and Prasher [38], La Fata, Lupo [39], Nag and Helal [30] | Healthcare and pharmacy industry |
| Sirisawat and Kiatcharoenpol [8], Prakash and Barua [37], Zarbakhshnia, Soleimani [35] | Logistics |
| Yüksel, Dağdeviren [40], Wu, Wei [41], D'Urso, Disegna [42], Atsalakis, Atsalaki [43], Büyüközkan, Feyzioğlu [44] | Tourism, hospitality |
| Pak, Thai [11], Sayareh, Iranshahi [14], Pantouvakis [17], Hemalatha, Dumpala [5] | Container port industry |
| Celik and Akyuz [45], Dožić, Lutovac [34], Rezaeenour [33] | Transportation towards airline, railroad, ship |
| Samanlioglu and Ayağ [32], Sharma, Gupta [46], Nojavan, Heidary [31] | Training and education |
| Ecer [47], Ji, Zhang [48], Li and Sun [49] | Banking, e-commerce, website design |

The research of Ugboma, Ibe [16] on the impact of PSQ on users' satisfaction in developing countries showed that "responsiveness" and "tangibles" dimensions of PSQ received the highest responses from customer's viewpoint, whereas "empathy" dimension had the lowest ratings. From these results, the study suggested that the port company should focus on the dimensions having the lowest ratings. Specifically, provision of service should assure punctuality; staff have to express willingness to support customers' requirements. These suggestion is consistent with that of Hemalatha, Dumpala [5], who posit that the ability to understand and share the feelings of customer will result in consumers' behavioral intentions, then leading to repurchase intentions and the word of mouth in the future.

By applying structural equation modeling (SEM), the study of Thai [56] confirmed that PSQ is a construct including four dimensions, namely process, management, outcomes, and image and social responsibility. Also, PSQ has a significant relationship with customer satisfaction. In line with previous studies, Sayareh, Iranshahi [14] proved that "reliability", "tangibles", "responsiveness", "empathy", and "assurance" are the determinants of PSQ, which significantly affect customer satisfaction. Further, "tangibles" is judged as the most important dimension among them. This result is in line with the research of Hemalatha, Dumpala [5].

On the purpose of identifying the quality of service provided by the international container ports in Asia from carriers' perspectives, Chou and Ding [50] used integrated the MCDM-IPA approach. Results show that PCs ought to grow the number of the port of call, build more import/export containers, and reduce port costs to enhance service quality and improve competitive capability. Further, many limited resources are being allocated to operations, such as port facilities and equipment, should be employed elsewhere, including transshipment container attraction. The role of reallocating resources, namely capital and human resources, is also discussed by Hu and Lee [57]. The results from the novel 3D model revealed that some service attributes should be progressed as soon as possible, for instance, port congestion, service promises, settlement of accident claims, and port users' requirements. Moreover, terminal operators are interested in ports employing high technologies in operations, such as artificial intelligence and block chain [56]. Because the application of modern technologies can help ports attract more shipping lines for the port of call [6].

Cho, Kim [58] demonstrated that PSQ is formed from three dimensions, namely endogenous quality, relational quality, and exogenous quality. Endogenous quality relates to internal capabilities of a port, consisting of loading and unloading charges, berthing facilities, and terminal capacity. Meanwhile, relational quality associates with relationships between PCs and shipping companies (SCs), including the port logistic network, employee professionalism, the customer partnership. Inversely, exogenous quality correlates with external factors influencing the magnetism of a port, including the port location, the cargo volume, the distance. For the Shanghai port, three dimensions of PSQ positively affect customer satisfaction and considerably differed among different customer groups [58]. Specifically, when compared among small and medium SCs, the bigger ones responded that PSQ is the crucial determinant of customer satisfaction and loyalty. This finding implies requirements for strategic investments to improve PSQ for larger SCs at both internal and external levels.

## 2.3. The IPA model

IPA was originally introduced by Martilla and James [59] to identify which SAs or products should pay more attention or which of them should be cut down the allocated resources. Additionally, IPA helps identify SAs that, first, are judged as the most important from the customer's perspective and definitely affect customer satisfaction the most [4, 60], and, second, have a low degree of satisfaction and need to be improved [4, 59].

**Fig 1. Importance-performance analysis grid.**

Traditionally, IPA is depicted by a two-dimension matrix classified into four parts (also called as quadrants), where importance attributes are represented along the horizontal axis while performance attributes are described along the vertical axis (Fig 1). SAs in Quadrant I describing as high importance and high performance represents for increasing a firm's competitive advantages, implying that the firm should "keep up the good work". SAs depicted in Quadrant II with characteristics of high importance, but low satisfaction, need to devote to immediate attention. A firm should concentrate on Quadrant II to increase the overall customer satisfaction. Ignorance of them may cause many serious threats. Attributes in Quadrant III have both low importance and low performance, thus known as "low priority" and thereby unnecessary to allocate additional resources here; whilst attributes in Quadrant IV representing as low in importance and high in satisfaction is known as "possible overkill", implying that resources spent to these attributes should be employed elsewhere.

On the one hand, the traditional IPA has been used in so many various settings, including tourism, healthcare, hotel and hospitality [3, 61–63]. On the other hand, other studies have been trying to modify this model to become feasible in the specific context [4, 5, 62, 64]. Although the modification of the model results in identifying better SAs needing to be improved, for example attributes in Quadrant II, they don't provide the improvement priorities in case of limited resources [65–67].

## 3. Method

### 3.1. The research framework

Fig 2 describes three main steps for the implementation of this research.

Step 1 is to identify the determinants of PSQ (also called SAs) and then set up the hierarchical structure of PSQ is created. To do so, we base on literature review and expert interview.

Step 2 is to adopt the fuzzy AHP to compute the original weights of importance and satisfaction attributes from port users' perspectives. Yet, the fuzzy AHP's necessary assumption is independence among attributes in its hierarchical structure [68]. Hence, instead of the conventional fuzzy AHP, we utilize the extension fuzzy AHP to adjust SAs' original weights.

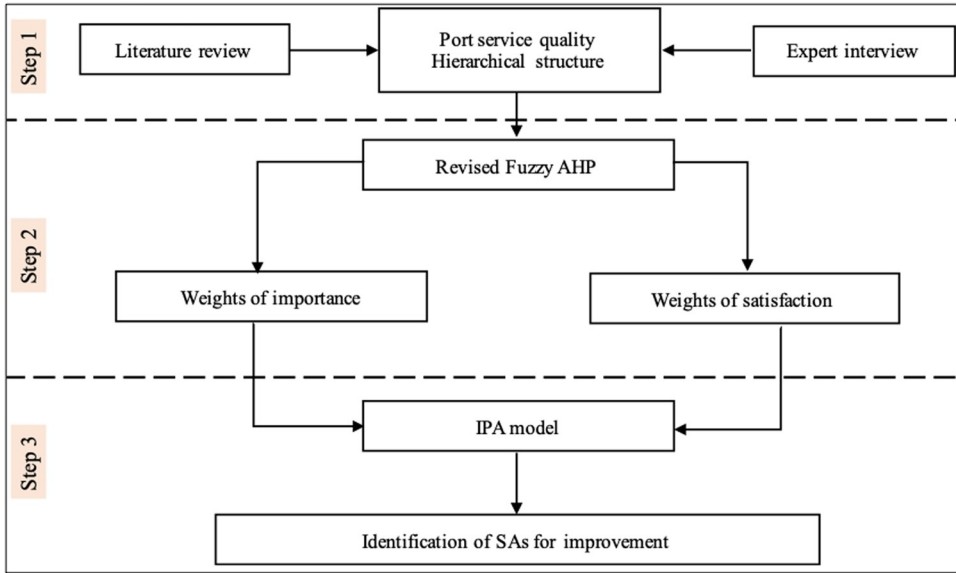

**Fig 2. The research framework.**

For Step 3, based on the results of the revised fuzzy AHP, the IPA model is employed to determine which SAs should be allocated to the limited resources. By which, some managerial implications are proposed to improve PSQ and satisfy customer demands.

## 3.2. The hierarchical structure of PSQ

Presently, SERVQUAL, which is originally proposed by Parasuraman, Zeithaml [69], is considered as one of the most popular measurement scales so as to measure service quality of the port industry [5, 10, 50, 57]. From the literature review, as show in Section 2.2, this article initially extracts 37 SAs for PSQ after excluding overlapping expression because one service attribute could have many ways to express. Next, this paper adopts the five-score judgment scale, ranging from 1 (very unclear) to 5 (very clear), to measure readability, accuracy, conciseness, and representativeness for SAs. To do so, the content authenticity index (CAI) is used to decide which SAs should be held for the next analysis: $CAI = (\sum_{i=1}^{5} C_i/5) \times 100\%$, with $C_i$ being the expert judgment of "4" and "5." Besides, respondents reached a consensus that SAs will be kept if their $CAI \geq 75\%$; apart from that, they will be deleted. As a results, 29 SAs satisfy $CAI \geq 75\%$ after three rounds of discussion, as exhibited in the 2$^{nd}$ field of Table 2.

Furthermore, expert interview and literature review were carried out to explore six dimensions of PSQ, including "tangibles", "reliability", "assurance", "empathy", "responsiveness", and "diversity" (Table 2). Although "diversity" is not initially embraced in the SERQUAL scale, through the results of Hsu, Yu [10] as well as the interview, the experts posited that "diversity" is a quite important dimension of PSQ that relates to provision of the different services for port users, for instance, logistics processing services, special cargo-related services, inland transportation, and value-added services.

## 3.3. Questionnaire design

This research aims to evaluate the PSQ by using the fuzzy AHP approach, hence the nine-point questionnaire proposed by Saaty [68], is utilized to weight the degree of importance and

**Table 2. The hierarchical structure of SAs.**

| Dimensions | Criteria | Code | Sources |
|---|---|---|---|
| Tangibles (TA) | Geographical location | TA1 | Cho, Kim [58], Hu and Lee [57], Pak, Thai [11] |
| | Cargo handling facilities and equipment | TA2 | Bae, Kim [3], Sayareh, Iranshahi [14], Hsu and Huang [65], expert interview |
| | Storage space for cargos | TA3 | Chou and Ding [50], Sayareh, Iranshahi [14], Thai [56] |
| | Berthing availability | TA4 | Hemalatha, Dumpala [5], Lee and Hu [15], Pantouvakis [17], Sayareh, Iranshahi [14] |
| | Information technology ability | TA5 | Chou and Ding [50], Hsu and Huang [65], expert interview |
| Reliability (RL) | Accuracy and consistency of schedules | RL1 | Cho, Kim [58], Hu and Lee [57], Pantouvakis [17] |
| | Detailed schedule | RL2 | Chou and Ding [50], Hemalatha, Dumpala [5], Pantouvakis [17], Hu and Lee [57] |
| | Accuracy of the bill of lading | RL3 | Hsu, Yu [10], expert interview |
| | Perfect transportation of cargos | RL4 | Hsu and Huang [65], Pak, Thai [11], Sayareh, Iranshahi [14], Thai [56] |
| Assurance (AS) | Efficient in handling customer complaints | AS1 | Hemalatha, Dumpala [5], Hsu, Yu [10], Ugboma, Ibe [16] |
| | Employees possess professional skills/knowledge | AS2 | Hemalatha, Dumpala [5], Hsu, Yu [10], Ugboma, Ibe [16], Sayareh, Iranshahi [14] |
| | Ability in dealing with cargo damage | AS3 | Chou and Ding [50], Hemalatha, Dumpala [5], Pantouvakis [17], Hu and Lee [57] |
| | Trustworthiness | AS4 | Chou and Ding [50], Hemalatha, Dumpala [5], Pantouvakis [17], Sayareh, Iranshahi [14] |
| | Comprehensive applications of ICT in customer service | AS5 | Chou and Ding [50], Hemalatha, Dumpala [5], Pantouvakis [17], Hu and Lee [57], Cho, Kim [58] |
| | Prompt responses of customer requirements | AS6 | Chou and Ding [50], Hemalatha, Dumpala [5], Pantouvakis [17], Hu and Lee [57], Sayareh, Iranshahi [14] |
| Empathy (EM) | Proactive provision of vessel schedules | EM1 | Hsu, Yu [10], expert interview |
| | Proactive provision of loading modes | EM2 | Cho, Kim [58], Hemalatha, Dumpala [5], Pantouvakis [17], Hu and Lee [57], Sayareh, Iranshahi [14] |
| | Proactive adjustment of operating procedures when customers request | EM3 | Cho, Kim [58], Hemalatha, Dumpala [5], Pantouvakis [17], Hu and Lee [57], Sayareh, Iranshahi [14], Hsu, Yu [10] |
| | Prompt announcement of any changing | EM4 | Chou and Ding [50], Hemalatha, Dumpala [5], Pantouvakis [17], Hu and Lee [57], Sayareh, Iranshahi [14] |
| | Emphasis on the safety of operations and work | EM5 | Hemalatha, Dumpala [5], Hsu, Yu [10], Ugboma, Ibe [16], Sayareh, Iranshahi [14], expert interview |
| Responsiveness (RP) | Uniform charges for all customers | RP1 | Cho, Kim [58], Hemalatha, Dumpala [5], Pantouvakis [17], [56], Hu and Lee [57], Sayareh, Iranshahi [14] |
| | In-time delivery | RP2 | Cho, Kim [58], Hemalatha, Dumpala [5], Pantouvakis [17], Hu and Lee [57], Sayareh, Iranshahi [14] |
| | Availability of kinds of pertinent services | RP3 | Thai [56], Chou and Ding [50], Hemalatha, Dumpala [5], Pantouvakis [17], Hu and Lee [57], Sayareh, Iranshahi [14] |
| | Willingness in helping customers | RP4 | Hemalatha, Dumpala [5], Sayareh, Iranshahi [14], Ugboma, Ibe [16], Sirisawat and Kiatcharoenpol [8], expert interview |
| Diversity (DI) | Provision of logistics processing services | DI1 | Thai [56], Hsu, Yu [10], Hemalatha, Dumpala [5], Pantouvakis [17], Hu and Lee [57], Sayareh, Iranshahi [14] |
| | Provision of special cargo-related services | DI2 | Thai [56], Hemalatha, Dumpala [5], Pantouvakis [17], Hu and Lee [57], Sayareh, Iranshahi [14], Chou and Ding [50] |
| | Provision of inland transportation | DI3 | Hsu, Yu [10], expert interview |
| | Diversification of service price | DI4 | Chou and Ding [50], Hu and Lee [57] |
| | Increase in value-added of a port user | DI5 | Expert interview |

satisfaction attributes of PSQ from port user's perspectives. The procedure for completing the survey questionnaire is as follow:

*Firstly*, the measurement scale of PSQ included six dimensions with 32 observed variables (also known as criteria). Then, one form of the questionnaire was drafted and pre-tested by seven practical employees (three from SCs, two from maritime freight forwarders, and two from the port company) to check if statements were easy for respondents to understand or whether important questions were missing.

*Secondly*, we modified the drafted questionnaire basing on the pre-testing results. Specifically, three confused statements were removed, and the other twelve questions were corrected to ensure concise and clear expressions.

*Finally*, the modified questionnaire was post-tested with the same number of subjects as in the above pre-test. As such, the official questionnaire consists of six dimensions with a total of 29 criteria, as mentioned in Table 2. Also, the questionnaire comprises two parts: Part 1 relates to general information of respondents while Part 2 correlates to the questions in terms of the degree of importance and satisfaction of PSQ.

### 3.4. Sampling

In the beginning, we intended to interview 20 experts from 11 SCs and ocean freight forwards. But only 19 agreed to join the interview. Thus, we directly interviewed the experts at their office and asked them to fill in the questionnaire. To assure the reliability of collected data, the respondents were opted based on two requirements: (1) the respondent had many years working in the import and export sector, (2) they were holding a managerial position at the workplace.

Because this study used a fuzzy AHP approach to compute weights of SAs, we only selected the answers that had a consistency index (CI) and the consistent ratio (CR) of less than 10% [68]. The CI and CR are symbolized as:

$$CI(n) = \frac{L_{\max} - n}{n - 1} \tag{1}$$

$$CR(n) = \frac{CI(n)}{MRCI(n)} \tag{2}$$

Where $L_{\max}$ is the maximum eigenvalue of the individual pair-wise comparison matrix (IPCM), which is formed by experts' judgments. And $n$ is the number of the criteria of each IPCM. Meanwhile, MRCI is a mean random consistency index, whose values are shown in Table 3. By adopting the package 'AHP survey' in the RStudio, only 15 out of 19 responses satisfied CR of less than 10%, meaning that these official 15 responses would be used for the next analysis.

As can be seen in Table 4, the majority of respondents have working experiences of greater than 11 years (67%). Further, all the subjects are holding the managerial position at their workplace, specifically the head of the division (40%), assistant manager (13.3%), vice manager (20%), and manager (26.7%). To conclude, the respondents' profile endorses the validity and reliability of the collected information.

### 3.5. The weights of PSQ

The weights of SAs include two parts, they are "local weights" and "global weights" [10, 50]. For simplification of explanation, this paper used the typical sample data of the RL dimension to explain in detail how to apply the extension fuzzy AHP approach in this research. The RL dimension in Table 2 includes 4 criteria: RL1, RL2, RL3 and RL4. Calculating two kinds of the weights of SAs by the extension fuzzy AHP approach was employed as follow:

**Table 3. MRCI values [70].**

| n | 3 | 4 | 5 | 6 | 7 | 8 | 9 | 10 |
|------|-------|------|------|------|------|------|------|------|
| MRCI | 0.525 | 0.89 | 1.11 | 1.25 | 1.35 | 1.40 | 1.45 | 1.49 |

**Table 4. Respondents' characteristics.**

|  | Characteristics | Frequency | % |
|---|---|---|---|
| Gender | Male | 13 | 86.7 |
|  | Female | 2 | 13.3 |
| Age in years | 25–30 | 1 | 6.7 |
|  | 31–40 | 4 | 26.7 |
|  | 41–40 | 7 | 46.7 |
|  | Above 50 | 3 | 20.0 |
| The educational level in years | Undergraduate | 5 | 33.3 |
|  | Master | 9 | 60.0 |
|  | Ph.D | 1 | 6.7 |
| Working experience (years) | 5–10 | 2 | 13.3 |
|  | 11–20 | 5 | 33.3 |
|  | 21–30 | 5 | 33.3 |
|  | Above 30 | 3 | 20.0 |
| Working position | Head of the Division | 6 | 40.0 |
|  | Assistant manager | 2 | 13.3 |
|  | Vice manager | 3 | 20.0 |
|  | Manager | 4 | 26.7 |
| Expertise | Financial management | 1 | 6.7 |
|  | Port operational management | 2 | 13.3 |
|  | Transportation control | 1 | 6.7 |
|  | Marketing logistics | 2 | 13.3 |
|  | Supply chain management | 4 | 26.7 |
|  | Others | 5 | 33.3 |

**3.5.1. Compute the fuzzy positive reciprocal matrix.**  In this study, the experts are asked to compare SAs using the Triangular Fuzzy Numbers (TFNs) and Triangular Fuzzy Reciprocal Numbers (TFRNs). TFNs are depicted in Fig 3.

The linguistic scale to measure SAs' importance level, as shown in Table 5, shows the relative magnitude of each dimension and criteria regarding each other and the corresponding

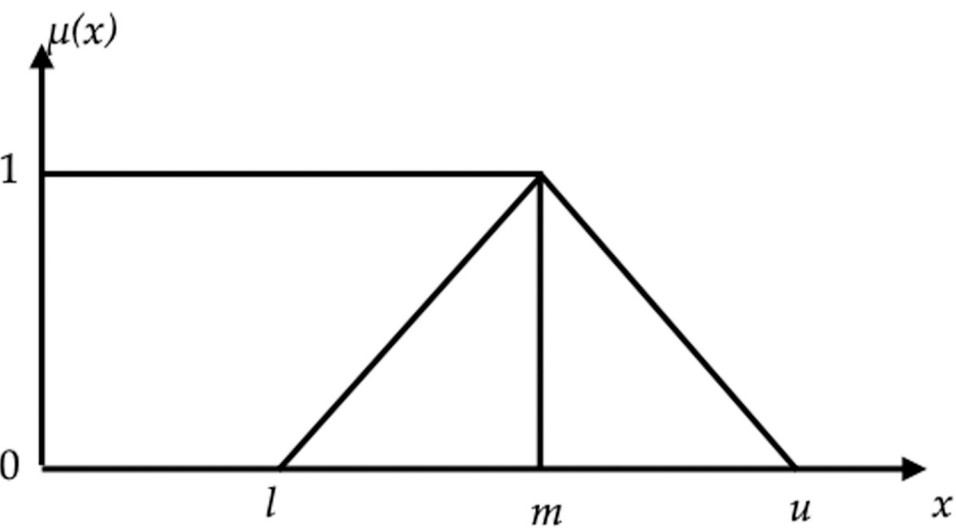

**Fig 3. A depiction of TFNs [71].**

Table 5. Linguistics measurement of importance scale [72].

| Degree of importance | Linguistic scale | Explanation | TFNs | TFRNs |
|---|---|---|---|---|
| 1 | Equally important (EI) | The importance of two SAs (A and B) is equal | (1, 1, 1) | (1, 1, 1) |
| 3 | Weakly more important (WI) | Judgement slightly favors A over B | (1, 3/2, 2) | (1/2, 2/3, 1) |
| 5 | Strongly more important (SI) | Judgement strongly favors A over B | (3/2, 2, 5/2) | (2/5, 1/2, 2/3) |
| 7 | Very strongly more important (VI) | An activity is preferred very strongly over another | (2, 5/2, 3) | (1/3, 2/5, 1/2) |
| 9 | Absolutely more important (AI) | The evidence favoring one activity over another is of the highest possible order of affirmation | (5/2, 3, 7/2) | (2/7, 1/3, 2/5) |

FTNs and TFRNs. If a respondent judge one SA to be strongly important than another, then FTNs are expressed as (3/2, 2, 5/2) and the other dimension will take (2/5, 1/2, 2/3) as TFRNs.

Next, the IPCM with $n$ SAs is established for the $k^{\text{th}}$ respondent. In this research, $k = 1,2,\ldots,15$.

$$\tilde{A}^{(k)} = \begin{vmatrix} \tilde{a}_{11}^k & \tilde{a}_{12}^k & \cdots & \tilde{a}_{1n}^k \\ \tilde{a}_{21}^k & \tilde{a}_{22}^k & \cdots & \tilde{a}_{2n}^k \\ \vdots & \vdots & \ddots & \vdots \\ \tilde{a}_{n1}^k & \tilde{a}_{n2}^k & \cdots & \tilde{a}_{nn}^k \end{vmatrix}$$

Then, 15 IPCMs are combined together by using the geometric mean into the fuzzy positive reciprocal matrix (FPRM), denoted $\tilde{A} = |\tilde{a}_{ij}|_{n \times n}$, where:

$$\tilde{a}_{ij} = \left[ \left( \prod_{k=1}^{15} l_{ij} \right)^{1/15}, \left( \prod_{k=1}^{15} m_{ij} \right)^{1/15}, \left( \prod_{k=1}^{15} u_{ij} \right)^{1/15} \right] = \left[ l_{ij}, m_{ij}, u_{ij} \right] \tag{3}$$

For the aforesaid RL dimension, the RL construct's fuzzy positive reciprocal matrix is:

$$\tilde{A}_{RL} = \begin{vmatrix} [1.000, 1.000, 1.000] & [0.948, 1.135, 1.354] & [1.133, 1.321, 1.525] & [0.923, 1.063, 1.221] \\ [0.739, 0.881, 1.055] & [1.000, 1.000, 1.000] & [1.411, 1.670, 1.931] & [1.145, 1.452, 1.757] \\ [0.656, 0.757, 0.883] & [0.518, 0.599, 0.709] & [1.000, 1.000, 1.000] & [0.622, 0.726, 0.861] \\ [0.819, 0.941, 1.084] & [0.569, 0.689, 0.874] & [1.162, 1.377, 1.607] & [1.000, 1.000, 1.000] \end{vmatrix}$$

**3.5.2. Test the consistency of FPRM.** FPRM is acceptable if its CR is less than 10% [68, 72, 73]. Yet, in this situation, we cannot compute CR as done in traditional AHP because inputs in FPRM are fuzzy numbers, not crisp numbers. Instead, we make use of a technique proposed by Kwong and Bai [73] to de-fuzzify the fuzzy numbers in FPRM into the crisp numbers. Then, CR can be calculated by the normal way of traditional AHP. According to Kwong and Bai [73], the fuzzy numbers $a_{ij} = [l_{ij}, m_{ij}, u_{ij}]$ may be de-fuzzified by the formula:

$$a_{ij} = \frac{l_{ij} + 4 \times m_{ij} + u_{ij}}{6}; i, j = 1, 2, \ldots, n. \tag{4}$$

We initially calculated FPRMs' $L_{\max}$ by using the package 'rARPACK' in the RStudio. After that, Formulas (1) and (2), as exhibited in Section 3.4, were carried out to estimate CR.

For the RL dimension, defuzzification of the matrix $\tilde{A}_{RL}$ is:

$$A_{RL} = \begin{vmatrix} 1.000 & 1.140 & 1.323 & 1.066 \\ 0.886 & 1.000 & 1.670 & 1.452 \\ 0.761 & 0.604 & 1.000 & 0.731 \\ 0.944 & 0.699 & 1.380 & 1.000 \end{vmatrix}$$

Then, the maximum eigenvalues of $A_{RL}$ may be estimated as $L_{max} = 4.037$, CI = 0.0122 and CR = 1.37% ($< 10\%$), thereby $A_{RL}$ is consistent. To sum up, the results of consistency tests for the remaining FPRMs demonstrated that all FPRMs are consistent because all their CR are less than 10% (Table 6).

### 3.5.3. The local weights SAs

In this research, we used a row geometric mean (RGM) to compute the local weights for each dimension and criterion. The process is carried out through 5 steps below:

*Step 1*: Let $\tilde{r}_i$ be the RGM vector, then the fuzzy evaluation matrix may be calculated as following [10]:

$$\tilde{r}_i = \left( \prod_{j=1}^{n} \tilde{a}_{ij} \right)^{1/n} = \left| \left( \prod_{j=1}^{n} l_{ij} \right)^{1/n}, \left( \prod_{j=1}^{n} m_{ij} \right)^{1/n}, \left( \prod_{j=1}^{n} u_{ij} \right)^{1/n} \right|, i = 1, 2, \ldots, n \qquad (5)$$

*Step 2*: Compute the sum of $\tilde{r}_i$ for each dimension and criterion, $\sum_{i=1}^{n} \tilde{r}_i$.

$$\sum_{i=1}^{n} (\tilde{r}_i) = \left[ \sum_{i=1}^{n} \left( \prod_{j=1}^{n} l_{ij} \right)^{1/n}, \sum_{i=1}^{n} \left( \prod_{j=1}^{n} m_{ij} \right)^{1/n}, \sum_{i=1}^{n} \left( \prod_{j=1}^{n} u_{ij} \right)^{1/n} \right] \qquad (6)$$

*Step 3*: Determine fuzzy weights by multiply each $\tilde{r}_i$ with the reverse FTNs obtained in

**Table 6. Consistency tests.**

| User's attributes | Dimension/Criteria | CI | RI | CR |
|---|---|---|---|---|
| Importance | Dimension | 0.0472 | 1.25 | 0.0378 |
| | Criteria 1: TA | 0.0515 | 1.11 | 0.0464 |
| | Criteria 2: RL | 0.0122 | 0.89 | 0.0137 |
| | Criteria 3: AS | 0.0117 | 1.25 | 0.0094 |
| | Criteria 4: EM | 0.0661 | 1.11 | 0.0595 |
| | Criteria 5: RP | 0.0134 | 0.89 | 0.0151 |
| | Criteria 6: DI | 0.0195 | 1.11 | 0.0176 |
| Satisfaction | Dimension | 0.0465 | 1.25 | 0.0372 |
| | Criteria 1: TA | 0.0570 | 1.11 | 0.0514 |
| | Criteria 2: RL | 0.0201 | 0.89 | 0.0226 |
| | Criteria 3: AS | 0.0191 | 1.25 | 0.0153 |
| | Criteria 4: EM | 0.0503 | 1.11 | 0.0453 |
| | Criteria 5: RP | 0.0165 | 0.89 | 0.0185 |
| | Criteria 6: DI | 0.0348 | 1.11 | 0.0314 |

Step 2:

$$\tilde{w}_i = \tilde{r}_i / \sum_{i=1}^{n} \tilde{r}_i = \left[ \frac{(\prod_{j=1}^{n} l_{ij})^{1/n}}{\sum_{i=1}^{n} (\prod_{j=1}^{n} u_{ij})^{1/n}}, \frac{(\prod_{j=1}^{n} m_{ij})^{1/n}}{\sum_{i=1}^{n} (\prod_{j=1}^{n} m_{ij})^{1/n}}, \frac{(\prod_{j=1}^{n} u_{ij})^{1/n}}{\sum_{i=1}^{n} (\prod_{j=1}^{n} l_{ij})^{1/n}} \right], i = 1, 2, \ldots, n \qquad (7)$$

*Step 4*: Compute the defuzzification of FTNs by the arithmetic mean method proposed by Kwong and Bai [73], as mentioned above. This step results in the unnormalized weights for SAs termed as ($M_i$).

*Step 5*: Normalize $M_i$ and then obtain a crisp local weight of the $i^{th}$ SAs by the formula:

$$N_i = \frac{M_i}{\sum_{i=1}^{n} M_i}; \ i = 1, 2, \ldots, n. \qquad (8)$$

For the RL dimension as an example, based on step 1, the fuzzy evaluation matrix may be found as:

$$\tilde{r}_i = \begin{vmatrix} 0.998 & 1.123 & 1.260 \\ 1.045 & 1.209 & 1.375 \\ 0.678 & 0.758 & 0.857 \\ 0.858 & 0.972 & 1.111 \end{vmatrix}$$

Applying Steps (2) and (3), the fuzzy weights for the $i^{th}$ RL ($i$ = 1,2,...4) as:

$$W_i = \begin{vmatrix} 0.217 & 0.277 & 0.352 \\ 0.227 & 0.298 & 0.384 \\ 0.147 & 0.186 & 0.239 \\ 0.186 & 0.239 & 0.310 \end{vmatrix}$$

Finally, by Steps (4) and (5), we have:

$$M_i = \begin{vmatrix} 0.2792 \\ 0.3003 \\ 0.1888 \\ 0.2423 \end{vmatrix} \quad => \quad N_i = \begin{vmatrix} 0.2763 \\ 0.2972 \\ 0.1868 \\ 0.2397 \end{vmatrix}$$

By the same way, as shown from Sections 3.5.1–3.5.3, the SAs' original weights can be obtained and exhibited in Table 7.

**3.5.4. The revising procedure of the SAs' original weights.** In theory, the AHP approach assumes that it exists the independence among criteria (dimensions) in the hierarchical structure [68, 72]. Yet, this assumption seldom satisfies in many real-world situations [10, 65]. To reflect the inter-effect among criteria in the hierarchical structure, this article adopts a direct-influential matrix to revise their original weights. The revision process is implemented via 4 steps:

(1) Forming the direct-influential matrix

**Table 7. The original weights for SAs.**

| Dimension | Global weights in the first-order (%) | | Criteria | Local weights in the second-order (%) | |
|---|---|---|---|---|---|
| | Importance weight | Satisfaction weight | | Importance weight | Satisfaction weight |
| Tangibles | 21.11 | 15.45 | TA1 | 23.12 | 31.23 |
| | | | TA2 | 34.23 | 11.34 |
| | | | TA3 | 16.23 | 23.45 |
| | | | TA4 | 12.09 | 23.06 |
| | | | TA5 | 14.33 | 10.92 |
| Reliability | 14.21 | 23.23 | RL1 | 27.63 | 30.65 |
| | | | RL2 | 29.72 | 22.34 |
| | | | RL3 | 18.68 | 15.45 |
| | | | RL4 | 23.97 | 31.56 |
| Assurance | 17.32 | 14.06 | AS1 | 20.09 | 12.34 |
| | | | AS2 | 30.99 | 19.45 |
| | | | AS3 | 23.43 | 23.44 |
| | | | AS4 | 11.00 | 10.12 |
| | | | AS5 | 6.34 | 23.09 |
| | | | AS6 | 8.15 | 11.56 |
| Empathy | 14.01 | 17.98 | EM1 | 11.23 | 9.34 |
| | | | EM2 | 17.47 | 16.47 |
| | | | EM3 | 23.56 | 21.23 |
| | | | EM4 | 34.11 | 18.89 |
| | | | EM5 | 13.63 | 34.07 |
| Responsiveness | 14.09 | 14.99 | RP1 | 21.56 | 34.12 |
| | | | RP2 | 31.45 | 24.76 |
| | | | RP3 | 16.98 | 20.91 |
| | | | RP4 | 30.01 | 20.21 |
| Diversity | 19.26 | 14.29 | DI1 | 34.11 | 13.24 |
| | | | DI2 | 21.23 | 17.34 |
| | | | DI3 | 19.01 | 23.67 |
| | | | DI4 | 17.19 | 21.38 |
| | | | DI5 | 8.46 | 24.37 |

Suppose that we have a direct-influential matrix D with $n$ SAs:

$$D = [d_{ij}]_{n \times n}; \ i,j = 1, 2, \ldots, n. \tag{9}$$

In the aforesaid equation, the $d_{ij}$ represents the inter-effect between the $i^{th}$ criterion and the $j^{th}$ criterion. Besides, the extent that a criterion impacting itself is not considered, implying that the $d_{ij} = 0$. This study deployed a 5-points Likert-scale, ranging from 1 = very low influence to 5 = very strong influence, to measure the inter-effect between the $i^{th}$ criterion and the $j^{th}$ criterion.

In our paper, seven practical experts among the 15 respondents, as seen in Section 3.4, was selected to determine values for $d_{ij}$ via a roundtable discussion. As a result, criteria's inter-effect in terms of the RL construct is shown in Fig 4. We can see that the direct-influential degree of RL1 on RL2 is 1.0 and that of RL2 on RL1 is 2.0. Thus, we have $d_{12} = 1.0$ and $d_{21} = 2.0$.

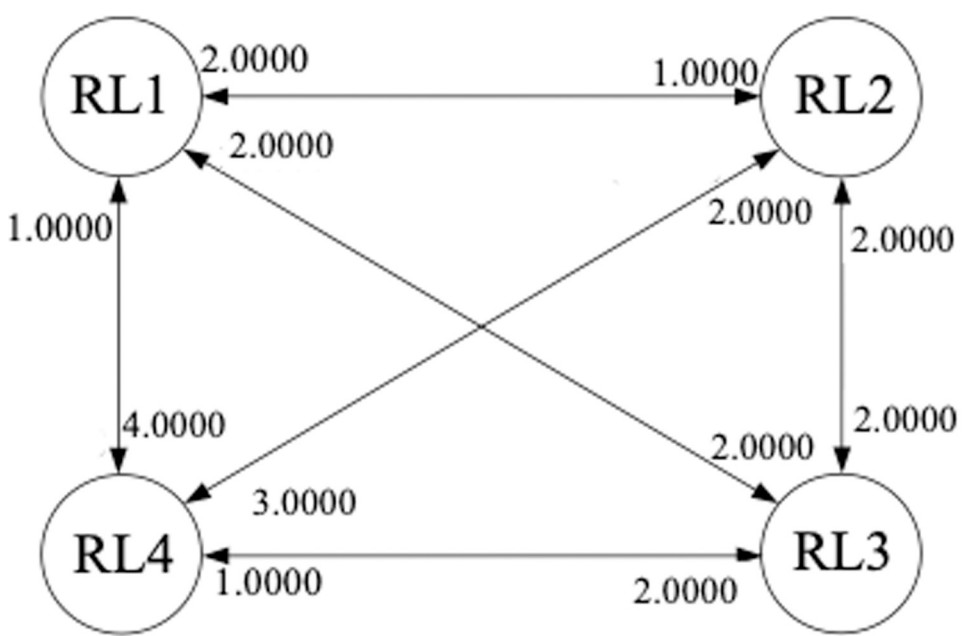

**Fig 4. The direct-influential matrix for the RL construct.**

Based on Fig 4 and Eq (9), the direct-influential matrix for the RL construct is attained as:

$$D = \begin{vmatrix} 0.0 & 1.0 & 2.0 & 4.0 \\ 2.0 & 0.0 & 2.0 & 3.0 \\ 1.0 & 2.0 & 0.0 & 1.0 \\ 1.0 & 2.0 & 2.0 & 0.0 \end{vmatrix}$$

(2) Normalizing the matrix D:

For the matrix D, the row-based sum $(\sum_{i=1}^{n} d_{ij})$ denotes for the total effects of the $i^{th}$ criterion on the others; thus, its maximum effect is defined by $\max_{1 \leq i \leq n} \sum_{j=1}^{n} r_{ij}$. Similarly, the column-based sum illustrates the total effects $j^{th}$ criterion on the others; hence, its maximum effects is obtained by $\max_{1 \leq j \leq n} \sum_{i=1}^{n} r_{ij}$. Let

$$G = Max\left[ \max_{1 \leq i \leq n} \sum_{j=1}^{n} d_{ij}, \quad \max_{1 \leq j \leq n} \sum_{i=1}^{n} d_{ij} \right] \qquad (10)$$

Next, the direct-influential matrix *is* normalized by:

$$P = \left[ \frac{d_{ij}}{G} \right]_{n \times n} ; \quad i, j = 1, 2, \ldots n. \qquad (11)$$

For the RL construct, from Eq (10), we have $G = 8.0$

Thus, based on Eq (11), the influential matrix $R$ can be normalized:

$$P = \begin{vmatrix} 0 & 0.1250 & 0.2500 & 0.5000 \\ 0.2500 & 0 & 0.2500 & 0.3750 \\ 0.1250 & 0.2500 & 0 & 0.1250 \\ 0.1250 & 0.2500 & 0.2500 & 0 \end{vmatrix}$$

(3) Normalizing the direct-influential matrix in long-run

In principle, when one criterion impacts another, then its impact will decrease gradually in long-run. In this circumstance, this paper defines the normalized direct-influential matrix in long term, as follows:

$$U = P + P^2 + \cdots + P^t, \ k \to \infty \tag{12}$$

By matrix operations, we have:

$$P \cdot U = P^2 + P^3 \cdots + P^t + P^{t+1}, \ t \to \infty \tag{13}$$

Subtracting (13) from (12), we have:

$$U(I - P) = P - P^{t+1} = P(I - P^t) \tag{14}$$

Since the value in matrix $P$ ranges from 0 to 1; thus, $\lim_{t \to \infty} P^t = O$. Therefore, when $t \to \infty$ (i.e. long-run), Eq (14) is rewritten as follows:

$$U(I - P) = P \quad => \quad U = \frac{P}{I - P} \tag{15}$$

For the RL construct, based on Eq (15), the matrix $U$ is achieved as:

$$U = \begin{vmatrix} 0.3722 & 0.6106 & 0.7479 & 1.0086 \\ 0.5901 & 0.5026 & 0.7616 & 0.9537 \\ 0.3705 & 0.5249 & 0.3619 & 0.5523 \\ 0.4117 & 0.5832 & 0.6244 & 0.5026 \end{vmatrix}$$

(4) Revising the SAs' original weight

The SAs' revised original weight includes two components: its original weight obtained by the conventional fuzzy AHP approach (Seeing Table 7), and the influential effects, which is computed by $U \times W$. Where $W$ is the vector of the original weight, as shown in Table 7. Let $W^R = [w_1^R, w_2^R, \ldots, w_n^R]$ represent SAs' revised weights vector. Then, we have:

$$W^R = W + U \times W \tag{16}$$

For the RL construct, based on Eq (16), the revised weight vector of SAs is calculated as:

$$W^R = \begin{vmatrix} 0.276 \\ 0.297 \\ 0.187 \\ 0.240 \end{vmatrix} + \begin{vmatrix} 0.3722 & 0.6106 & 0.7479 & 1.0086 \\ 0.5901 & 0.5026 & 0.7616 & 0.9537 \\ 0.3705 & 0.5249 & 0.3619 & 0.5523 \\ 0.4117 & 0.5832 & 0.6244 & 0.5026 \end{vmatrix} X \begin{vmatrix} 0.276 \\ 0.297 \\ 0.187 \\ 0.240 \end{vmatrix} = \begin{vmatrix} 0.9421 \\ 0.9805 \\ 0.6452 \\ 0.7639 \end{vmatrix}$$

Finally, we normalize the $W^R$ as:

$$\omega_i^n = \frac{W_i^R}{\sum\limits_{i=1}^{n} W_i^R} * 100\% \tag{17}$$

In the RL construct, the revised weight vector of the SAs is finally normalized as:

$$\omega_i^n = \begin{vmatrix} 0.2828 \\ 0.2943 \\ 0.1936 \\ 0.2293 \end{vmatrix}$$

Based on the above revised process, the SAs' revised weights in the other dimensions for importance measure can also be obtained, shown in the fourth column of Table 8. Likewise,

**Table 8. The revised weights of SAs for importance measurement (%).**

| Dimension | Global weight (A) | Criteria | Local weight (B) | Global weight (C = A x B) |
|---|---|---|---|---|
| Tangibles | 20.15 | TA1 | 21.75 | 4.38 |
| | | TA2 | 24.86 | 5.01 |
| | | TA3 | 16.50 | 3.32 |
| | | TA4 | 18.31 | 3.69 |
| | | TA5 | 18.58 | 3.74 |
| Reliability | 15.54 | RL1 | 28.28 | 4.39 |
| | | RL2 | 29.43 | 4.57 |
| | | RL3 | 19.36 | 3.01 |
| | | RL4 | 22.93 | 3.56 |
| Assurance | 15.97 | AS1 | 17.43 | 2.78 |
| | | AS2 | 16.75 | 2.67 |
| | | AS3 | 25.89 | 4.13 |
| | | AS4 | 13.65 | 2.18 |
| | | AS5 | 13.98 | 2.23 |
| | | AS6 | 12.30 | 1.96 |
| Empathy | 21.07 | EM1 | 26.13 | 5.51 |
| | | EM2 | 19.11 | 4.03 |
| | | EM3 | 17.70 | 3.73 |
| | | EM4 | 16.27 | 3.43 |
| | | EM5 | 20.79 | 4.38 |
| Responsiveness | 12.53 | RP1 | 26.13 | 3.27 |
| | | RP2 | 23.08 | 2.89 |
| | | RP3 | 17.51 | 2.19 |
| | | RP4 | 33.28 | 4.17 |
| Diversity | 14.74 | DI1 | 20.35 | 3.00 |
| | | DI2 | 29.12 | 4.29 |
| | | DI3 | 17.04 | 2.51 |
| | | DI4 | 13.86 | 2.04 |
| | | DI5 | 19.63 | 2.89 |

**Table 9. The revised weights of SAs for satisfaction measurement (%).**

| Dimension | Global weight (A) | Criteria | Local weight (B) | Global weight (C = A x B) |
|---|---|---|---|---|
| Tangibles | 20.96 | TA1 | 16.31 | 3.42 |
| | | TA2 | 24.24 | 5.08 |
| | | TA3 | 18.05 | 3.78 |
| | | TA4 | 16.55 | 3.47 |
| | | TA5 | 24.85 | 5.21 |
| Reliability | 16.36 | RL1 | 32.09 | 5.25 |
| | | RL2 | 31.08 | 5.08 |
| | | RL3 | 16.55 | 2.71 |
| | | RL4 | 20.28 | 3.32 |
| Assurance | 17.29 | AS1 | 21.53 | 3.72 |
| | | AS2 | 17.53 | 3.03 |
| | | AS3 | 16.78 | 2.9 |
| | | AS4 | 15.26 | 2.64 |
| | | AS5 | 14.22 | 2.46 |
| | | AS6 | 14.68 | 2.54 |
| Empathy | 19.93 | EM1 | 21.82 | 4.35 |
| | | EM2 | 23.71 | 4.73 |
| | | EM3 | 17.55 | 3.5 |
| | | EM4 | 19.01 | 3.79 |
| | | EM5 | 17.91 | 3.57 |
| Responsiveness | 13.18 | RP1 | 27.60 | 3.64 |
| | | RP2 | 29.20 | 3.85 |
| | | RP3 | 18.49 | 2.44 |
| | | RP4 | 24.71 | 3.26 |
| Diversity | 12.28 | DI1 | 21.87 | 2.69 |
| | | DI2 | 22.73 | 2.79 |
| | | DI3 | 17.74 | 2.18 |
| | | DI4 | 18.40 | 2.26 |
| | | DI5 | 19.26 | 2.37 |

the SAs' revised weights for dissatisfaction measure can be found in the fourth column of Table 9.

**3.5.5. SAs' global weights.** The SAs' global weights are calculated by multiplying their global weights in the first order by their revised local weights in the second order. Consequently, the SAs' global weights for importance measure and satisfaction measures are shown in the last column of Tables 8 and 9, respectively.

## 3.6. The importance-performance analysis

Based on the results in Tables 8 and 9 respectively, the global weights of importance and satisfaction are averaged approximately 3.45%, classified the quadrant matrix into four areas as be shown in Fig 5. Some managerial solutions for each PSQ are also proposed.

The results argued that there are nine SAs in Quadrant I with high expectation and high performance; thus, the policy for these SAs should "keep up the good work". Similarly, Quadrant II contained five SAs with high importance but low performance; so, the policy for these SAs should "concentrate here". In other words, port authorities should put more emphasis on these SAs, and more resources should be allocated on these. There are ten SAs in Quadrant III

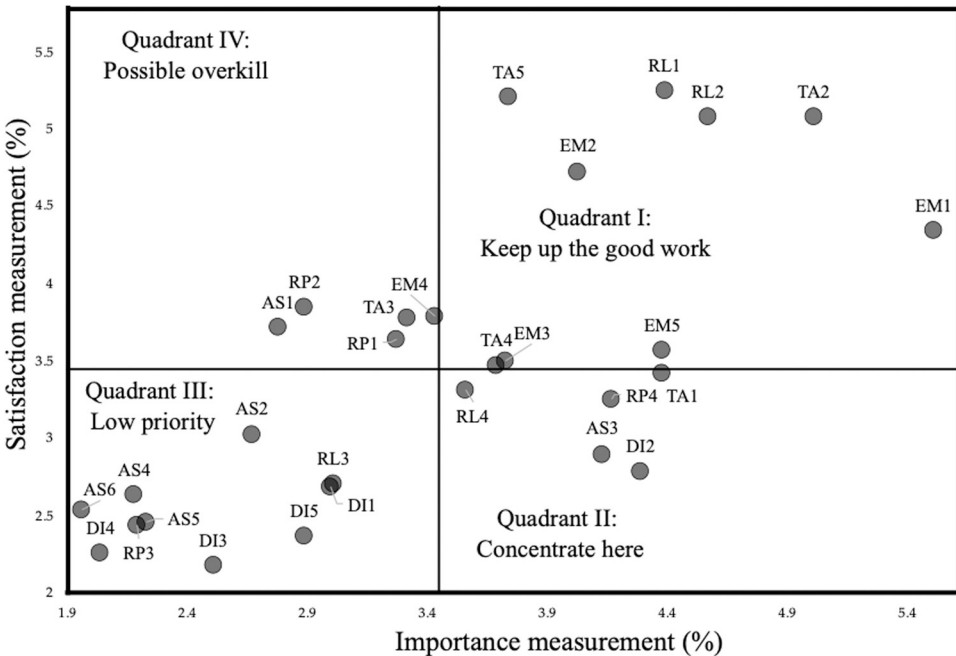

**Fig 5. The IPA results.**

with low importance and low performance. They are set as "low priority", implying that port managers do not need to put priorities on these SAs. It is noteworthy that three SAs with low importance but high satisfaction in Quadrant IV are evaluated as "overkill", signifying that limited resources allocating on these SAs should be reallocated elsewhere, especially transferred to SAs in Quadrant II.

# 4. Results, discussions, and managerial implications

## 4.1. Results and discussions

As to be listed in Table 8, among six dimensions of SAs, the port users put the most attention on "empathy" (21.07%) and "tangibles" (20.15%), while "responsiveness" received the least interests. The fact that "empathy" and "tangibles" considered as the top two dimensions from port users' viewpoint is understandable in the service sector generally and the port industry in particular. According to Pantouvakis, Chlomoudis [74], "empathy" could be understood as the port's capacity to inform its customers immediately of any problems regarding their transportation, including schedules, modes of transportation, transit cost. Besides, "empathy" affects the accuracy of transit, therefore influencing the business efficiency of port users, particularly shipping company [11, 12]. Also, "tangibles" reflect port equipment, facilities, as well as the instructions and information inside the port, consisting of the availability of the intermodal transport network [65], the magnitude of the terminal region [14], the number and availability of berths at the port [56], thereby having a great impact on a port selection from shipping carriers. This result is quite consistent with studies of Ugboma, Ibe [16] and Chou and Ding [50].

Likewise, among 29 criteria of SAs, there are top five criteria concerned the most by the shipping company, including EM 1 ("proactive provision of vessel schedules", 5.51%), TA2 (cargo handling facilities and equipment, 5.01%), RL2 ("detailed schedule", 4.57%), RL1 ("accuracy and consistency of schedules", 4.39%), TA1 ("geographical location", 4.38%). The above information is useful for port managers, as well as port authorities in proposing plausible

solutions to better SAs in the future. In the present study, we discovered that the management process in DNPC is divergent among divisions and departments. On the one hand, this will lead to difficulties in providing the high-quality service for customers, in the other hand, may translate into imprecision in the transaction contract between staff and customers. In practice, to improve the above SAs, port managers should apply some specific solutions, such as (1) standardization of managerial procedure by using ISO that is currently very popular on service industries, (2) focusing on training the working skills for front-line staff who directly work with customers, (3) diversification of provided services, especially special cargo-related services.

Considered the results from Fig 5, the port company should prioritize the investment on SAs in Quadrant II, including "perfect transportation of cargos", "ability in dealing with cargo damage", "willingness in helping customers", "provision of special cargo-related services". Furthermore, the results showed that the port company is allocating limited resources unnecessarily for SAs in quadrant IV. So, it is necessary to transfer scarce resources from SAs in Quadrant IV into SAs in Quadrant II in order to better port users' satisfaction.

### 4.2. Managerial implications

From a managerial standpoint, port authorities can find the following proposed solutions to be useful in improving PSQ and, in turn, improving customer satisfaction.

First of all, it is recommended that port managers concentrate on enhancing and expanding the existing port infrastructure system, as well as pay more attention to canals dredging in order to pick up transportation capacity and surmount intra-port traffic jams, which happens more and more frequently, especially during the high seasons. The next important point is to expand warehousing facilities to satisfy demand for port logistical activities. This suggestion is relatively consistent with that of Li, Lan [9]. Besides that, port executives should increase awareness among their employees about the importance of a customer-oriented culture, as well as provide them with the necessary skills and behaviors. Hu and Lee [57] also place an emphasis on setting up a uniform code of conduct for the provision of port services to spread customer-centric culture throughout the port company. Meanwhile, Chou and Ding [50] highlight the significance of strengthening foreign languages skills for all staff because international logistics operations requires employees to understand documents which are written in foreign languages, for instance, English or Chinese. Surveyed experts also suggested that port managers should bolster the port reliability by applying advanced port management practices to the whole inland areas, including container yards, maintenance facilities, and warehouses. It is argued that this solution could improves the efficiencies of port operations [58]. Last but not least, procedures of customs clearance and processes of goods delivery/receipt should be simplified to save time and costs for customers. Huo, Zhang [12], Notteboom, Parola [7], and Hsu, Huang [22] likewise have the similar suggestions.

### 5. Conclusions

Thank to the ongoing growth of economic activities over two decades, the port industry has been playing a crucial role in the Vietnamese national economy. To attract more and more SCs to use port services, port managers and port authorities must know which factors affect the port users' expectation and perception. That is why the improvement of SAs become a basic part in the recent port development strategy of the Vietnamese government. The research towards PSQ using both F-AHP and IPA, to the best of our knowledge, has not yet conducted in Vietnam before. Thus, this study aims to investigate PSQ by using the F-AHP approach, and the IPA model from port users' perspectives. This article may also provide the valuable contributions for further research regarding PSQ using both F-AHP and IPA.

From the relevant literature and expert interview, the hierarchical structure of PSQ embracing six dimensions with 29 criteria was initially established. To test the research model, DNPC at Dong Nai port and their service users were empirically investigated. The results prove that the importance degree of dimensions is ranked as follow: empathy (21.07%), tangibles (20.15%), assurance (15.97%), reliability (15.54%), responsiveness (12.53%), diversity (14.74%). Meanwhile, for criteria of PSQ, top five criteria concerned the most by SCs include including "proactive provision of vessel schedules", "cargo handling facilities and equipment", "detailed schedule", "accuracy and consistency of schedules", and "geographical location".

Results from the IPA model show that four SAs in Quadrant II needing to prioritize for improvement, namely "perfect transportation of cargos", "ability in dealing with cargo damage", "willingness in helping customers", "provision of special cargo-related services". The practical policy is that port authorities should transfer the limited resources from SAs in Quadrant IV to SAs in quadrant II to enhance PSQ and attract more SCs and freight forwarders.

Although our research provides a lot of practical and theoretical references for the port industry of Vietnam, there are several limitations needing to be considered. *First of all*, this study only collected raw data by interviewing respondents from two kinds of port users, for instance shipping company and ocean freight forwarders. Therefore, further research should extend the respondents from other forms of port users, for example container terminal company, port tourists, and logistics companies inside the port, to better the robustness of the findings. *Secondly*, because of the utilization of cross-sectional data, this research cannot analyze the changing trend of PSQ over time. Hence, other scholars should carry out a longitudinal study so as to exactly assess insights the development of PSQ over a period of time.

## Supporting information

**S1 File.**
(ZIP)

## Acknowledgments

The authors would like to thank colleagues for very thoughtful reviews and critical comments, which have led to significant improvements to the early versions of the manuscript.

## Author Contributions

**Conceptualization:** Thang Quyet Nguyen, Nguyen Tan Huynh.

**Data curation:** Lan Thi Tuyet Ngo, Thanh Le Quoc, Long Van Hoang.

**Formal analysis:** Lan Thi Tuyet Ngo, Long Van Hoang.

**Investigation:** Nguyen Tan Huynh, Long Van Hoang.

**Methodology:** Lan Thi Tuyet Ngo.

**Project administration:** Thang Quyet Nguyen.

**Software:** Nguyen Tan Huynh, Thanh Le Quoc.

**Supervision:** Thang Quyet Nguyen.

**Writing – original draft:** Lan Thi Tuyet Ngo, Nguyen Tan Huynh.

**Writing – review & editing:** Thanh Le Quoc, Long Van Hoang.

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
