## [Decision Letter · Decision Letter 0]

7 Dec 2021

PONE-D-21-32601Assessing port service quality: An application of the extension Fuzzy AHP and Importance-Performance AnalysisPLOS ONE

Dear Dr. Huynh,

Thank you for submitting your manuscript to PLOS ONE. After careful consideration, we feel that it has merit but does not fully meet PLOS ONE’s publication criteria as it currently stands. Therefore, we invite you to submit a revised version of the manuscript that addresses the points raised during the review process.

We look forward to receiving your revised manuscript.

Kind regards,

Mehdi Keshavarz-Ghorabaee

Academic Editor

PLOS ONE

Journal Requirements:

Reviewers' comments:

Reviewer's Responses to Questions

5. Review Comments to the Author

Reviewer #1: Thank you for inviting me to review this interesting article. I realized that it is a very important study for assessing port service quality. However, the following points need to be improved before publication.

1. Rationale of the topic and more detailed results of the present work should be given in the abstract section.

2. Why triangular fuzzy numbers are preferred should be emphasized. The rationale for the methods used must be clearly stated in the introduction sections of the study. Because there are a lot of MCDM algorithm in the literature. Try to refer more recent MCDM articles-related (i.e., port, logistics service, 3PL, etc.), for this purpose, the following article are suggested:

10.1109/ACCESS.2021.3121607

10.3390/axioms10020048

10.3390/axioms10010034

10.3390/math9080886

3. Why 29 criteria are considered should be explained?

4. Providing more information about experts would be better.

5. Which aggregation operator are used while combining expert judgments?

6. It is recommended to write the managerial impact section.

7. The manuscript requires minor revision by native speakers.

Reviewer #2: This study aims to investigate port service quality, then an real study case is presented. The paper is quite well organised and the contribution is quite well positioned in the current background.

The introduction section should more in depth clarify that they refer to container ports however the methodology can be used also for other types of port. For example, can it be used for short-sea shipping one?

The authors are suggested to detail the method proposed giving some useful info also for readers not so experts as they are.

Fig.2's description needs to follow closer the framework proposed, clarifying each steps pictured. Besides, the description of methods (subsections of section 3) could follow the research framework presented in Fig. 2. For example, each subsection title could be equal to box in Figure 2.

Formulas are not legible, e.g. eq. 2 and 3, 4 and so on. Symbols are also not legible in the main text (e.g. see row 294).

---

## [Author Response · Author response to Decision Letter 0]

3 Feb 2022

I already inserted the file named "Response to reviewers" in your system.

---

## [Editor Report · Decision Letter 1]

14 Feb 2022

Assessing port service quality: An application of the extension Fuzzy AHP and Importance-Performance Analysis

PONE-D-21-32601R1

Dear Dr. Huynh,

We’re pleased to inform you that your manuscript has been judged scientifically suitable for publication and will be formally accepted for publication once it meets all outstanding technical requirements.

Kind regards,

Mehdi Keshavarz-Ghorabaee

Academic Editor

PLOS ONE

---

## [Editor Report · Acceptance letter]

16 Feb 2022

PONE-D-21-32601R1 

Assessing port service quality: An application of the extension Fuzzy AHP and Importance-Performance Analysis 

Dear Dr. Huynh:

I'm pleased to inform you that your manuscript has been deemed suitable for publication in PLOS ONE. Congratulations! Your manuscript is now with our production department. 

Kind regards, 

on behalf of

Dr. Mehdi Keshavarz-Ghorabaee 

Academic Editor

PLOS ONE